# Effects of OH radical and SO₂ concentrations on photochemical reactions of mixed anthropogenic organic gases

Junling Li[1], Kun Li[2,3], Hao Zhang[1], Xin Zhang[1], Yuanyuan Ji[1], Wanghui Chu[1], Yuxue Kong[1], Yangxi Chu[1], Yanqin Ren[1], Yujie Zhang[1], Haijie Zhang[1], Rui Gao[1], Zhenhai Wu[1], Fang Bi[1], Xuan Chen[1], Xuezhong Wang[1], Weigang Wang[4], Hong Li[1,*], Maofa Ge[4,*]

[1] State Key Laboratory of Environmental Criteria and Risk Assessment, Chinese Research Academy of Environmental Sciences, Beijing 100012, China

[2] Laboratory of Atmospheric Chemistry, Paul Scherrer Institute, 5232 Villigen, Switzerland

[3] Environment Research Institute, Shandong University, Qingdao 266237, China

[4] State Key Laboratory for Structural Chemistry of Unstable and Stable Species Beijing National Laboratory for Molecular Sciences (BNLMS), CAS Research/Education Center for Excellence in Molecular Sciences, Institute of Chemistry, Chinese Academy of Sciences, Beijing 100190, China

*Correspondence to*: Hong Li (lihong@craes.org.cn), Maofa Ge (gemaofa@iccas.ac.cn)

**Abstract.** Vehicle exhaust, as a major source of air pollutants in urban areas, contains a complex mixture of organic vapors including long-chain alkanes and aromatic hydrocarbons. The atmospheric oxidation of vehicle emissions is a highly complex system as the co-existing inorganic gases (e.g., $NO_x$ and $SO_2$) from other urban sources, and therefore remains poorly understood. In this work, the photooxidation of *n*-dodecane, 1,3,5-trimethylbeneze, and their mixture are studied in the presence of $NO_x$ and $SO_2$ to mimic the atmospheric oxidation of urban vehicle emissions (including diesel and gasoline vehicles), and the formation of ozone and secondary aerosols are investigated. It is found that ozone formation is enhanced by higher OH concentration and higher temperature, but is influenced little by $SO_2$ concentration. However, $SO_2$ can largely enhance the particle formation in both number and mass concentrations, likely due to the promoted new particle formation and acid-catalyzed heterogeneous reactions from the formation of sulfuric acid. In addition, organo-sulfates and organo-nitrates are detected in the formed particles, and the presence of $SO_2$ can promote the formation of organo-sulfates. These results provide a scientific basis for systematically evaluating the effects of $SO_2$, OH concentration, and temperature on the oxidation of mixed organic gases in the atmosphere that produce ozone and secondary particles.

## 1 Introduction

Atmospheric fine particulate matter with diameters < 2.5 μm ($PM_{2.5}$) is a common air pollutant that has a variety of adverse health outcomes (Requia et al., 2018; Tsai et al., 2013; Crouse et al., 2012). Organic aerosol (OA) is an important type of $PM_{2.5}$, and secondary organic aerosol (SOA) accounts for more than 50% of OA by mass concentration (S. Guo et al., 2010; Huang et al., 2014; Jimenez et al., 2009; Kanakidou et al., 2005). In China, concentrations of $PM_{2.5}$ have declined with the implementation of stringent emission control measures since 2013 (Gao et al., 2020; Zhang et al., 2019; Cheng et al., 2019).

The level of primary organic aerosol (POA) in $PM_{2.5}$ has been greatly reduced, however, the contribution of SOA to $PM_{2.5}$ has increased (Ming et al., 2017), highlighting the increasing importance of research on SOA.

Intermediate volatile organic compounds (IVOCs) have been found to contribute to a large fraction of SOA in both field observations (Fang et al., 2021; Wang et al., 2020a; Xu et al., 2020) and laboratory studies (Srivastava et al., 2022; Hu et al., 2021; Cai et al., 2019; Li et al., 2019a; Li et al., 2019b). Long-chain alkanes, as representatives of IVOCs, their laboratory studies are mainly focused on the case of a single long-chain alkanes or mixture of various precursors which include long-chain alkanes. Studies of single long-chain alkanes (e.g., *n*-decane, *n*-undecane, *n*-dodecane, *n*-tridecane, 2,6,10-trimethyl dodecane, nonyl-cyclohexane) include reaction kinetics (Lamkaddam et al., 2019; Shi et al., 2019a; Shi et al., 2019b), reaction mechanism (Li et al., 2020; Li et al., 2021a; Li et al., 2017b), analysis of gas phase and particle phase products (Fahnestock et al., 2015; Lamkaddam et al., 2020), quantification of particle yield (Docherty et al., 2021; Loza et al., 2014), and particle physicochemical properties (Li et al., 2017b; Li et al., 2020; Li et al., 2021a). For the mixture of various precursors which include long-chain alkanes, studies about which mainly focus on the chemical composition of the mixture gases, the properties of total organic carbon, the amount of SOA generated, and the effect of semi/intermediate volatile organic compounds (S/IVOCs) on the formation of SOA contribution (Qi et al., 2021; Hu et al., 2021; Qi et al., 2019; Cai et al., 2019; Deng et al., 2017; Li et al., 2021d; Li et al., 2019a; Li et al., 2019b). However, laboratory studies on the mixture of long-chain alkanes and aromatic hydrocarbons (e.g., 1,3,5-trimethylbeneze, m-xylene, benzene, toluene, ethylbenzene) are very limited (Li et al., 2021b), despite both of them being important SOA precursors in vehicle exhaust emissions (Qi et al., 2021).

As an important chemical component of inorganic pollutants in China, $SO_2$ has a high concentration in the urban atmosphere (Chu et al., 2016; Liu et al., 2016; Liu et al., 2017; Wang et al., 2019). Field observations in North China Plain showed that during heavy haze pollution episodes, $SO_2$ concentration could be >100 ppb, and the formation and growth rates of SOA and sulfate were much faster than that during clean periods (Li et al., 2017a). Laboratory studies demonstrated that the presence of $SO_2$ could enhance the SOA formation from anthropogenic and biogenic precursors, e.g., monoterpenes, isoprene, aromatics (Liggio and Li, 2013; Santiago et al., 2012; Tadeusz E. Kleindienst et al., 2006; Zhang et al., 2020; Yang et al., 2020; Liu et al., 2019). In addition, the presence of $SO_2$ could affect the light scattering and absorption properties of formed SOA (Zhang et al., 2020; Jaoui et al., 2008; Nakayama et al., 2015; Nakayama et al., 2018). It should be noted that most of the previous studies focused on the effect of $SO_2$ on the particle formation from single precursor. However, the studies on the impact of $SO_2$ on particle formation in mixture systems are very limited, although it has important atmospheric implications in better understanding the complex chemical processes in urban areas.

According to the field observation in China, higher concentration of 1,3,5-TMB and *n*-dodecane were observed, the 1,3,5-TMB concentration at rural site could reach 1.447 ppb, and the measured concentration of $C_{12}$ alkanes at rural site was 0.122±0.12 ppb (Chen et al., 2020; Wang et al., 2020a). In addition, the content of 1,3,5-TMB and n-dodecane in liquid gasoline cannot be ignored (Schauer et al., 2002; Gentner et al., 2012). In this work, a large outdoor smog chamber was applied to investigate the effects of $SO_2$ on particle formation from the mixture of *n*-dodecane and 1,3,5-trimethylbeneze

(1,3,5-TMB) in the presence of $NO_x$. Ozone and particle formation were analyzed. The results in this work are helpful to improve our understanding of the effect of inorganic gases on anthropogenic mixture organic compounds.

## 2 Experimental Section

### 2.1 Smog chamber experimental conditions

The experiments were performed in a 56 $m^3$ outdoor smog chamber, which was built on the rooftop of a building located at Chinese Research Academy Environmental Sciences (CRAES). The details of the chamber have been described elsewhere (Li et al., 2021c). Briefly, fluorinated ethylene propylene Teflon film (FEP 100, DuPont USA) was used as the reactor wall. Sunlight was the natural light source, and a $J_{NO2}$ filter radiometer (Metcon, Germany) was used to detect the irradiation intensity inside the chamber. The variation of temperature (T) and relative humidity (RH) inside the chamber were detected

by a temperature and humidity sensor (Beijing Star Sensor Technology Co., LTD.). Three fans were located on the opposite corner of the bottom of the chamber, which were used to mix the gas compounds and seed particles sufficiently. Before each experiment, the chamber was flushed with zero air for at least 24 hours with a flow rate of 200 L $min^{-1}$. A schematic of the experimental setup is shown in Figure S1.

  All the experiments in this work were performed in winter, of which the initial conditions and results were summarized

in Table S1 and S2. The entire photochemical reaction process for the conducted experiments lasted 7 hours; the enclosure was opened between 9:00-10:00 in the morning and closed at 16:30-17:30 in the afternoon. Temperature inside the chamber at noon was around 15-30 ℃. As the bottom of the reactor was made of aluminum, after a period of sunlight exposure, the surface temperature of the aluminum plate will rise. The chamber covered by Teflon film is equivalent to a greenhouse, the internal temperature will rise after the sunlight exposure. The cooling system of the chamber is water-cooled, in order to

prevent the cooling pipes from being frozen and cracked, the system is closed. Thus, the temperature inside the chamber during winter is higher than the ambient environment. The relative humidity during the whole photochemical process was <15%.

  The gas-phase *n*-dodecane and/or 1,3,5-TMB was introduced into the chamber by zero air with a known volume of liquid *n*-dodecane and/or 1,3,5-TMB, and the injector was heated gently during the sample injection process. $NO_x$ was used

as the OH precursor, NO and HONO experiments were designed, as their pathways of which for generating OH radicals in the atmosphere were different. HONO could directly generate OH radicals by photolysis; while for NO experiments, the generation of OH radical was through recycling via NOx/HOx chemistry (Ng et al., 2007). For high-$NO_x$ experiments, NO was introduced from a 500 ppm standard gas cylinder (500 ppm NO in nitrogen); for HONO experiments, HONO was prepared by the dropwise addition of 1 mL 5 wt% $NaNO_2$ into 2 mL 30 wt% $H_2SO_4$ in a glass bubbler, and the formed NO,

$NO_2$, and HONO was flushed into the chamber with zero air. The measured initial $NO_x$ concentration in the chamber was in the range of 315~445 ppb. $SO_2$ was introduced from a 60 ppm standard gas cylinder (600 ppm $SO_2$ in nitrogen). For experiments with low $SO_2$ concentration (L-HONO/NO-experiments), initial $SO_2$ concentration was in the range of 0~9.5

ppb; for experiments with high $SO_2$ concentration (H-HONO/NO-experiments), initial $SO_2$ concentration was in the range of 25.5~106 ppb. When the target species introduced into the chamber were mixed evenly, the enclosure of the chamber was open and the reaction started.

## 2.2 Online and offline measurements

Gaseous $NO_x$, $SO_2$, and $O_3$ concentration inside the chamber were monitored in real time by an $SO_2$ analyzer (EC 9850, Ecotech, Australia), an $O_3$ analyzer (EC 9830, Ecotech, Australia), and a $NO_x$ analyzer (EC 9841, Ecotech, Australia), respectively. A HONO analyzer (Chen et al., 2020) (Beijing Zhichen Technology Co., Ltd.) was used to measure the HONO concentration during the reaction process. Organic precursors in the chamber were collected with the Tenax TA sorbent before and after the photochemical reactions, and were then analyzed with a thermal desorption–gas chromatography with flame ionization detection (GC, 8890; TD, UNITY-xr). As the concentration of organic precursors after the photooxidation was nearly zero, so the initial concentrations were shown in Table S1 and S2 to represents the organic precursors consumed in the reaction.

The formed particles were monitored with a scanning mobility particle sizer (SMPS, Model 3080, Model 3081, and Model 3772, TSI Inc., USA). The particles were also collected with a low-flow sampler (LV 40BW, Sibata Scientific Technology Ltd., Soka, Japan) at a flow rate of 15 L min$^{-1}$ for 20 min with PTFE filters (0.2 μm, 47 mm, Merck Millipore, type FGLP). Then the collected whole PTFE filter was extracted with 5 mL methanol in an ultrasonic bath (KH5200DV, Hechuang Ultrasonic, China) for 30 min. The extracted solutions were analyzed with electrospray ionization quadrupole time-of flight mass spectrometry (ESI-Q-ToF-MS, Bruker Compact). Positive ion mode was used for the ESI-Q-ToF-MS, and the mass resolution of this instrument was > 20000.The concentration of inorganic species in aerosols and gases for mixture experiments in the chamber were measured with a Monitor for AeRosols and Gases in Ambient air (MARGA 2080, Applikon, Metrohm). The measured inorganic species in gases was nitric acid, and the inorganic species measured in aerosols included sulfate and nitrate. The attenuated total internal reflection infrared (ATR-IR) analysis was used to measure the potential functional groups in filter extracts; an FTIR spectrometer (Bruker, Tensor 27) equipped with a RT-DLaTGs detector was applied.

## 2.3 Calculation methods of SOA yields and OH concentration

Details of the calculation methods of wall-loss corrections and secondary aerosol (SA) yields can be referred to Li et al. (2021b). Briefly, when calculating the SA yields, the organic vapor and aerosol wall-loss corrections were both considered (Zhang et al., 2014). The ratio of average gas-particle partitioning timescale ($\bar{\tau}_{g-p}$) to the vapor wall-loss timescale ($\bar{\tau}_{g-w}$) could be used to evaluate the organic vapor wall-loss correction (Chen et al., 2019).

$\bar{\tau}_{g-p}$ can be expressed as the following equation:

$$\bar{\tau}_{g-p} = \frac{1}{2\pi \bar{N}_p \bar{D}_p D_{gas} \bar{F}_{FS}} \tag{1}$$

where $\bar{N}_p$ was the average number concentration of the formed particles during the experiment, $\bar{D}_p$ was the number mean diameter of the particles, $D_{gas}$ was the gas-phase diffusivity, $\bar{F}_{FS}$ was the Fuchs-Sutugin correction for noncontinuum mass transfer (Seinfeld J.H., 2016).

The gas-phase diffusivity $D_{gas}$ can be expressed as the following equation:

$$D_{gas} = D_{CO_2} \times \frac{M_{wCO_2}}{M_w} \tag{2}$$

where $D_{CO2}$ was $1.38 \times 10^{-5}$ m$^2$ s$^{-1}$, $M_w$ was set to 300 g mol$^{-1}$ here.

And the Fuchs-Sutugin correction for noncontinuum mass transfer $\bar{F}_{FS}$ can be expressed as following:

$$\bar{F}_{FS} = \frac{0.75\alpha(1+k_n)}{k_n^2 + k_n + 0.283k_n\alpha + 0.75\alpha} \tag{3}$$

where $\alpha$ was the mass accommodation coefficient onto particles, and it was set to 0.002 in this work (Zhang et al., 2014).

$$k_n = \frac{\lambda}{R_p} = \frac{6D_{gas}}{D_p\bar{c}} \tag{4}$$

where $K_n$ was the Knudsen number, $R_p$ was the particle radius, and $\lambda$ was the gas mean free path.

The vapor wall-loss timescale ($\bar{\tau}_{g-w}$) can be expressed as following:

$$\bar{\tau}_{g-w} = \frac{1}{k_w} \tag{5}$$

$$k_w = \left(\frac{A}{V}\right) \frac{a_w \frac{\bar{c}}{4}}{1.0 + \frac{\pi}{2}\left[\frac{a_w\bar{c}}{4(k_eD_{gas})^{0.5}}\right]} \tag{6}$$

where $k_w$ was the wall loss rates of the organic vapor; $\frac{A}{V}$ was the ratio of surface to volume of the chamber, 1.55 m$^{-1}$ for this chamber; $a_w$ was the mass accommodation coefficient of vapors deposition to the wall ($10^{-5}$ was used here) (Zhang et al., 2014); $\bar{c}$ was the root mean square speed of the gas; $k_e$ was the eddy diffusion coefficient, which was set to 0.12 s$^{-1}$ according to the reported values for a 60 m$^3$ chamber (McMurry and Grosjean, 1985).

$$\bar{c} = \sqrt{\frac{8RT}{\pi M_w}} \tag{7}$$

where $R$ was the ideal gas constant (i.e., 8.314 J mol$^{-1}$ K$^{-1}$), $T$ was the temperature, $M_w$ was the molecular weight.

The particle wall-loss was corrected based on the size-dependent coefficients from inert particle (ammonium sulfate) wall-loss experiments:

$$k_{dep}(d) = 6.35 \times 10^{-6}d^{1.56} + \frac{6.38}{d^{0.67}} \tag{8}$$

where $k_{dep}(d)$ was the wall-loss loss coefficient of particles in the diameter $d$.

In this work, OH was determined by measuring the concentration of tracer by TD-GC during the mixture experiments (Barmet et al., 2012). Changes in the tracer concentration over time can be expressed as:

$$\frac{d[tracer]}{dt} = -k[OH][tracer] \tag{9}$$

where $k$ is the rate constant for the reaction of tracer and OH radical. In the case of constant OH radical concentration level, equation (9) can be integrated to equation (10):

$$\ln[tracer]_0 = k[OH]t + \ln[tracer]_t \tag{10}$$

Plotting the natural logarithm (ln) of the tracer versus time (t),the slope that equals to $k$[OH] is obtained. Therefore, average OH radical concentration during each period is expressed as:

$$[OH] = \frac{\ln\frac{[tracer]_0}{[tracer]_t}}{kt} = \frac{slope}{k} \tag{11}$$

In this work, 1,3,5-trimethylbenzene is chosen as the tracer for the mixture experiments, as it reacts mainly with OH and has no interference from other compounds. The rate constants (Atkinson and Arey, 2003) at 298 K for reaction of 1,3,5-trimethylbenzene with OH radical is $5.67 \times 10^{-11}$ cm$^3$ molecule$^{-1}$ s$^{-1}$. Then the OH concentration is calculated with equation (11). Combined with the sampling frequency, the time resolution for [OH] calculations is about 1-1.5 hour.

The corresponding OH exposure is quantified by normalizing the 1,3,5-TMB concentration before the sampling period to the 1,3,5-TMB concentration before next sampling period and applying the known OH+1,3,5-TMB rate constant (Atkinson and Arey, 2003), as shown in Equation (12):

$$OH\ exposure = \frac{\ln\frac{[1,3,5-TMB]_0}{[1.3.5-TMB]_t}}{k_{OH+1,3,5-TMB}} \tag{12}$$

## 3 Results and Discussions

### 3.1 General results of the experiments

The HONO experiments were conducted as follows: 1,3,5-TMB + HONO + SO$_2$ (HONO-TMB), $n$-dodecane + HONO + SO$_2$ (HONO-Dod), 1,3,5-TMB + $n$-dodecane + HONO + SO$_2$ (HONO-Mix). The concentration of organic precursor was 137.9~216.9 ppb for 1,3,5-TMB and 23.2~28.9 ppb for $n$-dodecane. The measured NO$_x$ concentration applied in HONO experiments was in the range of 315~429 ppb. According to Ng et al. (2007), this method could generate $(6.3~8.6) \times 10^6$ molecules cm$^{-3}$ OH initially. As shown in Figure 1a, the concentration of OH radicals generated at the beginning of the experiment is in the range of $(1.03~1.23) \times 10^7$ molecules cm$^{-3}$, which is slightly higher than that of Ng et al. (2007). The OH exposure was in the range of $3.74 \times 10^{10}$ to $7.16 \times 10^{10}$ molecules cm$^{-3}$ s, as revealed in Figure 1b, corresponding to 6.9~13.3 simulated hours, assuming a global average OH concentration of $1.5 \times 10^6$ molecules cm$^{-3}$ (Mao et al., 2009). The reaction profiles of the HONO experiments are shown in Figure S2.

The NO experiments were conducted as follows: 1,3,5-TMB + NO + SO$_2$ (NO-TMB), $n$-dodecane + NO + SO$_2$ (NO-Dod), 1,3,5-TMB + $n$-dodecane + NO + SO$_2$ (NO-Mix). The concentration of organic precursor was 177.8~192.4 ppb for 1,3,5-TMB and 23~29.9 ppb for $n$-dodecane. The initial NO$_x$ concentration in the chamber was in the range of 212~355 ppb, resulting in the estimated OH concentration of $(3.4~4.9) \times 10^6$ molecules cm$^{-3}$, as shown in Figure 1. Using the OH concentration above, the calculated photochemical age from these experiments was in the range of 0.9~11.9 hours. The reaction profiles of the NO experiments are shown in Figure S3.

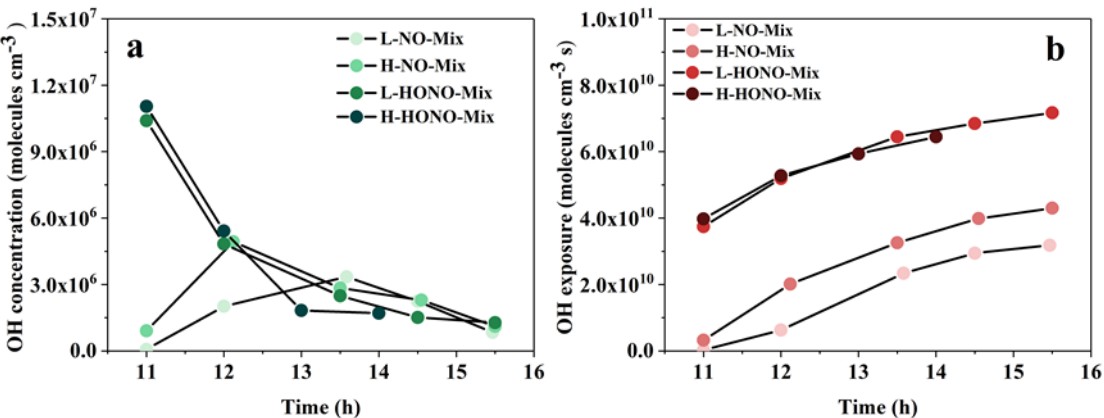

Figure 1. (a) OH radical concentration and (b) OH exposure versus time (hour) for mixture experiments. The points in this figure means the average OH concentration (Figure 1a) and OH exposure (Figure 1b) in the time interval between this data point and the previous data point. For the first experimental points in Figure 1, they are the average OH concentration (Figure 1a) and OH exposure (Figure 1b) in the time interval between 11:00 a.m. and the opening time of the photochemical reactions.

### 3.2 Ozone formation and gas phases products

### 3.2.1 Ozone formation

The ozone formation in the NO and HONO experiments are shown in Figure S2 and S3. In order to conduct a specific analysis, the highest concentration of ozone generated by each reaction is selected and shown in Figure 2 and Figure S4. It can be clearly seen that the addition of $SO_2$ has little effect on ozone generation, and the ozone generation was analyzed below from the perspective of the type of precursors, VOCs/$NO_x$, temperature, and the type of oxidant.

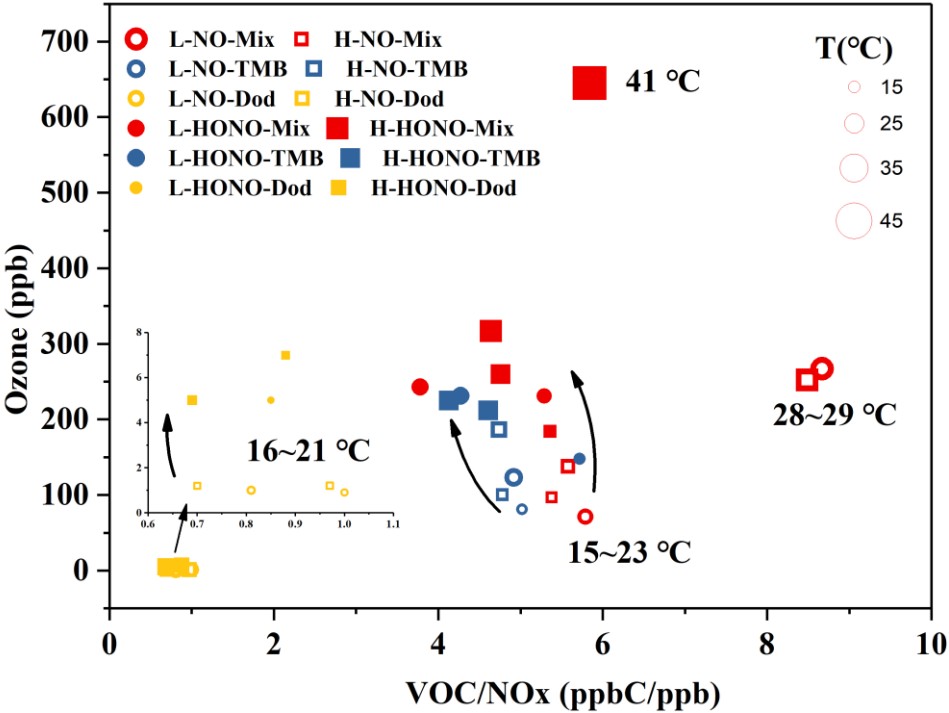

**Figure 2. Ozone formation in the NO and HONO experiments. The temperature (T) and ozone concentration here refers to the maximum value during the reaction process. The yellow circles refer to NO-Dod experiments with low SO₂ concentration, the yellow boxes refer to NO-Dod experiments with high SO₂ concentration; the yellow filled circles refer to HONO-Dod experiments with low SO₂ concentration, and the yellow filled boxes refer to HONO-Dod experiments with high SO₂ concentration. The blue circles refer to NO-TMB experiments with low SO₂ concentration, the blue boxes refer to NO-TMB experiments with high SO₂ concentration; the blue filled circles refer to HONO-TMB experiments with low SO₂ concentration, and the blue filled boxes refer to HONO-TMB experiments with high SO₂ concentration. The red circles refer to NO-Mix experiments with low SO₂ concentration, the red boxes refer to NO-Mix experiments with high SO₂ concentration; the red filled circles refer to HONO-Mix experiments with low SO₂ concentration, and the red filled boxes refer to HONO-Mix experiments with high SO₂ concentration.**

According to previous studies, the photochemical ozone formation potentials (OFP) of VOCs are sensitive to their rate constants with OH radicals, i.e., VOCs with high reactivities have greater contributions to the ozone formation in the ambient atmosphere (Jenkin and Hayman, 1999). The reaction rate constants with OH at 298 K for 1,3,5-TMB and *n*-dodecane are $5.67 \times 10^{-11}$ cm$^3$ molecule$^{-1}$ s$^{-1}$ and $1.39 \times 10^{-11}$ cm$^3$ molecule$^{-1}$ s$^{-1}$, respectively (Sivaramakrishnan and Michael, 2009; Atkinson and Arey, 2003). As shown in Figure 2, compared with the mixture and 1,3,5-TMB reaction systems, the ozone concentration generated by the *n*-dodecane system is very low (< 8 ppb). For *n*-dodecane experiments, the formed ozone concentration under HONO condition, the light pink area, is higher than that under NO condition, the pink area; for 1,3,5-TMB experiments, the ozone concentration under HONO condition in the light grey area is higher than that under NO conditions; the mixture experiments also show the similar phenomenon, the ozone concentration in light purple area is higher

than that in purple area. This can also be explained by the OH exposure: as shown in Figure 1, for similar $NO_x$ concentration, HONO conditions have higher OH radicals than NO experiments. Higher OH radicals would make the reaction system more
oxidative, forming more $RO_2$ and $HO_2$ that can react with NO. This leads to the competition to the reaction $O_3 + NO \rightarrow NO_2 + O_2$, and causes the accumulation of ozone. It is also shown in Figure 2 that the VOC to $NO_x$ ratio (within the range of 1~10 ppbC/ppb) has little effect on the generation of ozone. However, temperature has a great influence on the formation of ozone. For experiment H-HONO-Mix-3, the temperature at noon was 41 ℃, with a maximum ozone concentration of 645 ppb. In contrast, for the mixture experiments under similar VOC/NOx conditions but lower temperature (16~27℃), the ozone
concentration was 184~317 ppb. The reaction rates with OH increase with the rise of temperature (Atkinson and Arey, 2003), which in turn accelerate the oxidation processes. This may be a possible reason for the higher concentration of ozone generated under higher temperature conditions.

Figure 3 shows the concentration-time profiles of measured and simulated ozone and ozone formation and loss rates in H-HONO-Mix-4 experiment. The experiment was simulated with Master Chemical Mechanism MCM version3.3
(http://mcm.leeds.ac.uk/MCM/). The model was constrained with measured NO, $NO_2$, and HONO concentration. As shown in Figure 3a, the ozone production is well represented by the model in the first 0.5 h, however the model starts to over-predict the $O_3$ concentration in the after 5.5 h. This phenomenon is similar to a study about 1,3,5-TMB, the experiment of which was performed with an outdoor chamber (Metzger et al., 2008). Meanwhile, the ozone photochemical budget cycle was simulated with MCM model. The two major reactions of $HO_2$ + NO and $RO_2$ + NO control the photochemical
generation of ozone; $NO_2 + OH$, $NO_2 + RO_2$, and VOCs + $NO_3$ reactions control the ozone consumption.

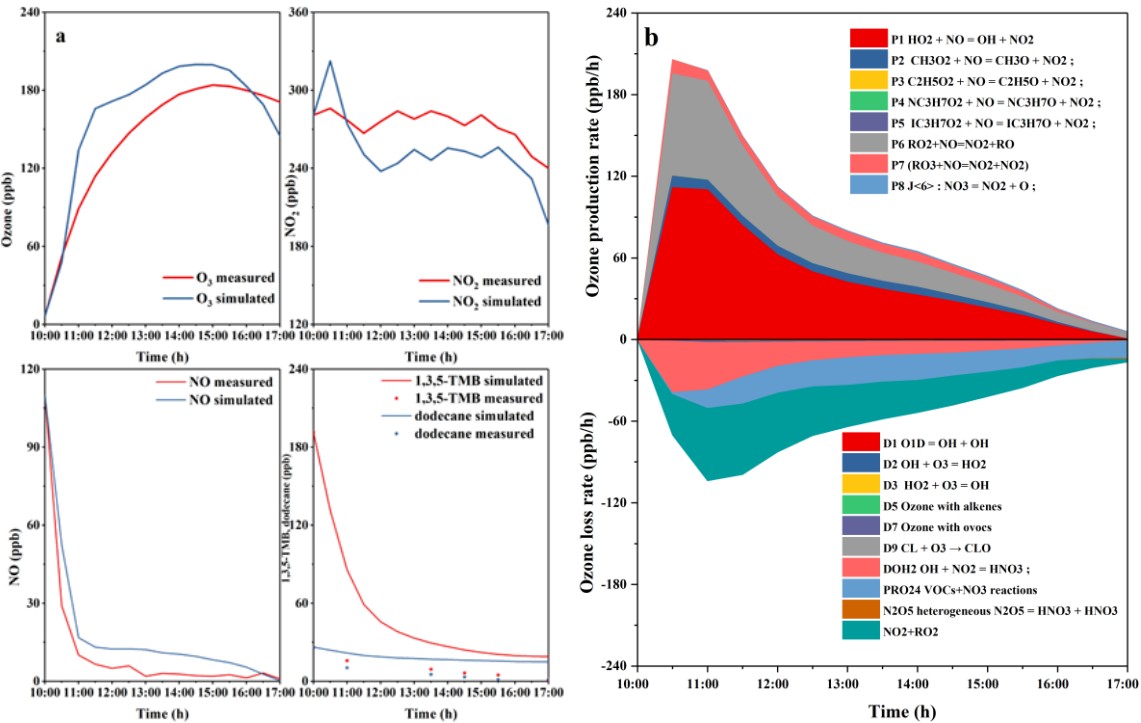

Figure 3. Concentration-time profiles of monitored and simulated (a) ozone, NO, NO$_2$, and 1,3,5-TMB and dodecane, and (b) ozone formation and loss rates in the H-HONO-Mix-4 experiment.

### 3.2.2 HONO concentration

For HONO experiments, HONO was introduced into the chamber with the same operation procedure, leading to similar HONO concentration. As shown in Figure 4a, initial HONO concentrations exceed the device detection limit (70 ppb). When the enclosure was open, the HONO concentration decreased rapidly with the progress of the reaction in the first two hours.

     For NO experiments, the concentration of HONO first slowly increased at about 1 h after the photochemical reaction started, and then decreased (Figure 4b). At the same time, the particles began to increase significantly after 1 hour to 2 hours
after the start of the reaction, as revealed in Figure S5. The increase of HONO concentration in NO experiments may have two reasons: the gas-phase reaction of NO with OH radicals and the heterogeneous reactions of NO$_2$ (Alicke, 2002; Wall and Harris, 2016). The concentrations of HONO generated in the NO-Mix and NO-1,3,5-TMB experiments are slightly lower than those in the NO-Dod experiments, and the NO-Mix experiments have the lowest HONO concentration. This is likely due to the difference in OH reactivity: the OH reactivity in the $n$-dodecane experiments (7.8-10.2 s$^{-1}$) is much lower than that
in the 1,3,5-TMB (255-267.4 s$^{-1}$) and mixture experiments (255.9-278.6 s$^{-1}$), leading to weaker competition to OH+NO reaction.

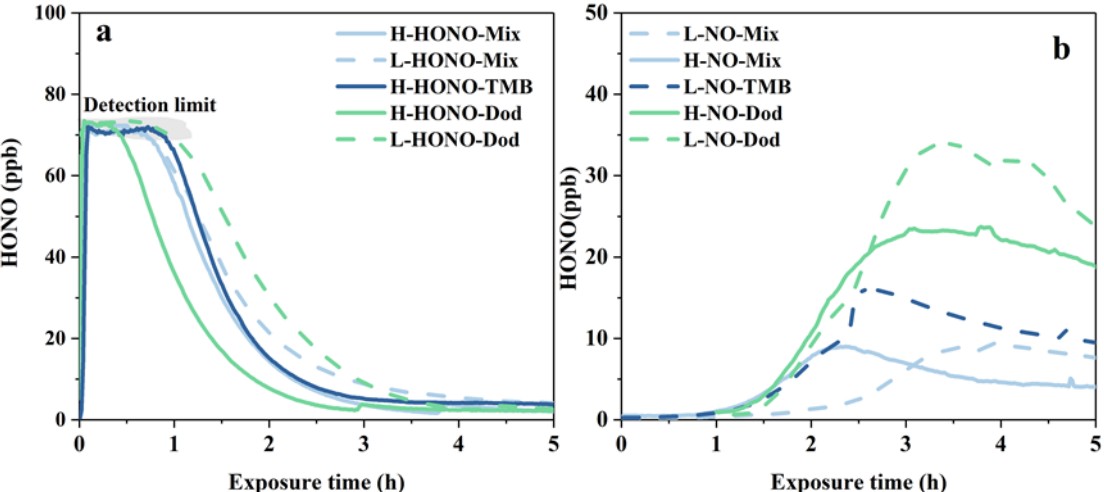

Figure 4. HONO concentration versus time during conducted experiments.

### 3.3 Effect of NO$_x$ and SO$_2$ on particle formation

The particle formation (size distribution and max number concentration) in the NO and HONO experiments are shown in Figures S5 and S6, and the corresponding max mass concentrations are shown in Figure 5 and Figure S7. When the max number concentrations are above $10^5$ #/cm$^3$, the banana like-formation particles occur, especially in experiments with high SO$_2$ concentration. It is shown that the higher SO$_2$ concentration can promote particle production in both number (Figures S5

and S6) and mass (Figure 5) concentration. This finding is similar to previous studies with single precursors (Liu et al., 2016; Liu et al., 2019), and is likely due to the formation of sulfuric acid from $SO_2$ oxidation. First, sulfuric acid can participate in nucleation and enhance the new particle formation (Sipila et al., 2010), resulting in higher particle number concentration. Second, sulfuric acid can promote the acid-catalyzed heterogeneous reactions and enhance the uptake of reactive organic compounds (Liu et al., 2016; Jang et al., 2002; Cao and Jang, 2007), which may lead to higher particle mass concentration. At last, the presence of $SO_2$ and sulfuric acid favour the formation of organo-sulfates (Liu et al., 2019; Liu et al., 2017; Chu et al., 2016), which is detected in our experiments (see Section 3.4).

In addition, it is found that the maximum values of particle number and mass concentration in the HONO reaction systems are higher than that of the NO reaction systems. In other words, under similar $NO_x$ and $SO_2$ concentrations, HONO conditions would be beneficial to the formation of particles. This phenomenon can be explained by higher OH exposure in HONO experiments, as shown in Figure 1b. Higher OH exposure causes the higher consumption rate of the precursors and the subsequent faster particle generation rates.

Figure 5 also shows the temperature effect on particle formation. For the same precursor, under the similar VOC/NOx conditions, the lower the temperature, the higher the mass of particulate matter. Lower temperatures can affect the partitioning process of organic vapor and facilitate the formation of particles, which in turn increases the particle mass concentration in the reaction system, and this is consistent with previous experimental reports (Ding et al., 2017; Li et al., 2020; Sheehan and Bowman, 2001; Takekawa et al., 2003; Warren et al., 2009). Meanwhile, with the increase of the VOC/$NO_x$ ratio value, the particle mass concentration increases (Wang et al., 2020b; Li et al., 2017c).

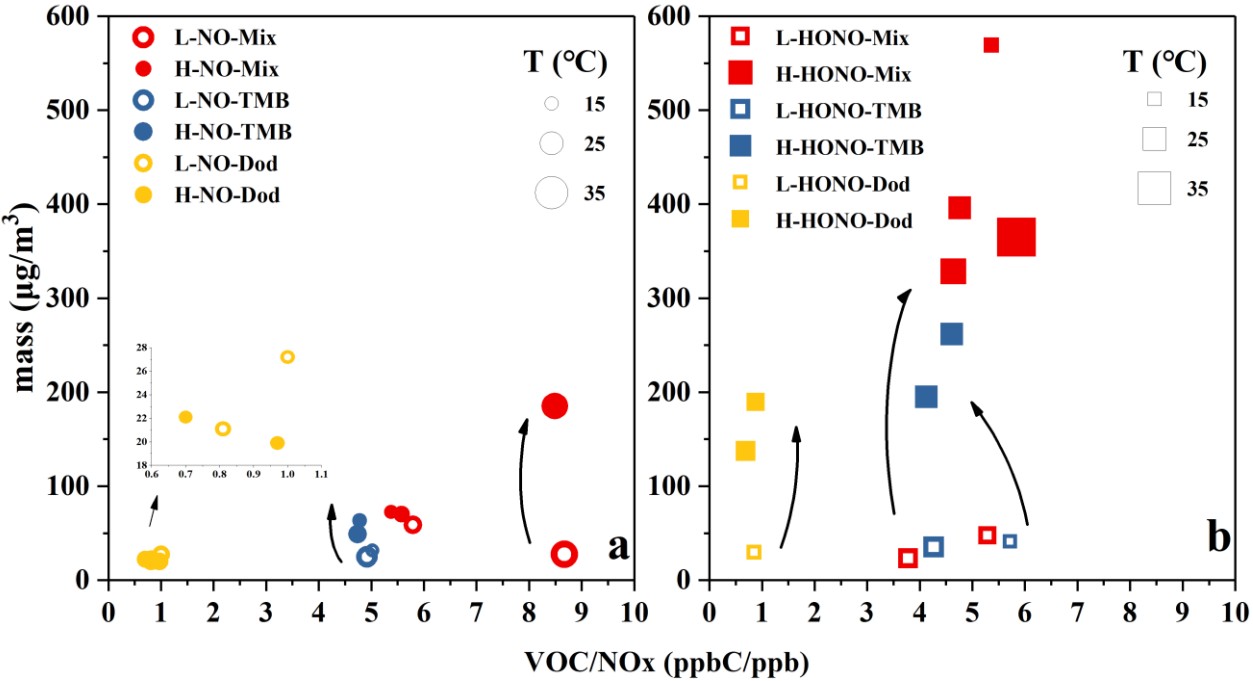

**Figure 5. Particle formation of NO and HONO experiments. The temperature (T) and particle mass concentration here refers to the maximum value during the reaction process. The yellow circles refer to NO-Dod experiments with low SO₂ concentration, the yellow filled circles refer to NO-Dod experiments with high SO₂ concentration; the blue circles refer to NO-TMB experiments with low SO₂ concentration, the blue filled circles refer to NO-TMB experiments with high SO₂ concentration; the red circles refer to NO-Mix experiments with low SO₂ concentration, the red filled circles refer to NO-Mix experiments with high SO₂ concentration. The yellow boxes refer to HONO-Dod experiments with low SO₂ concentration, the yellow filled boxes refer to HONO-Dod experiments with high SO₂ concentration; the blue boxes refer to HONO-TMB experiments with low SO₂ concentration, and the blue filled boxes refer to HONO-TMB experiments with high SO₂ concentration; the red boxes refer to HONO-Mix experiments with low SO₂ concentration, and the red filled boxes refer to HONO-Mix experiments with high SO₂ concentration.**

## 3.4 Chemical Compositions of Particles

### 3.4.1 Inorganic chemical components

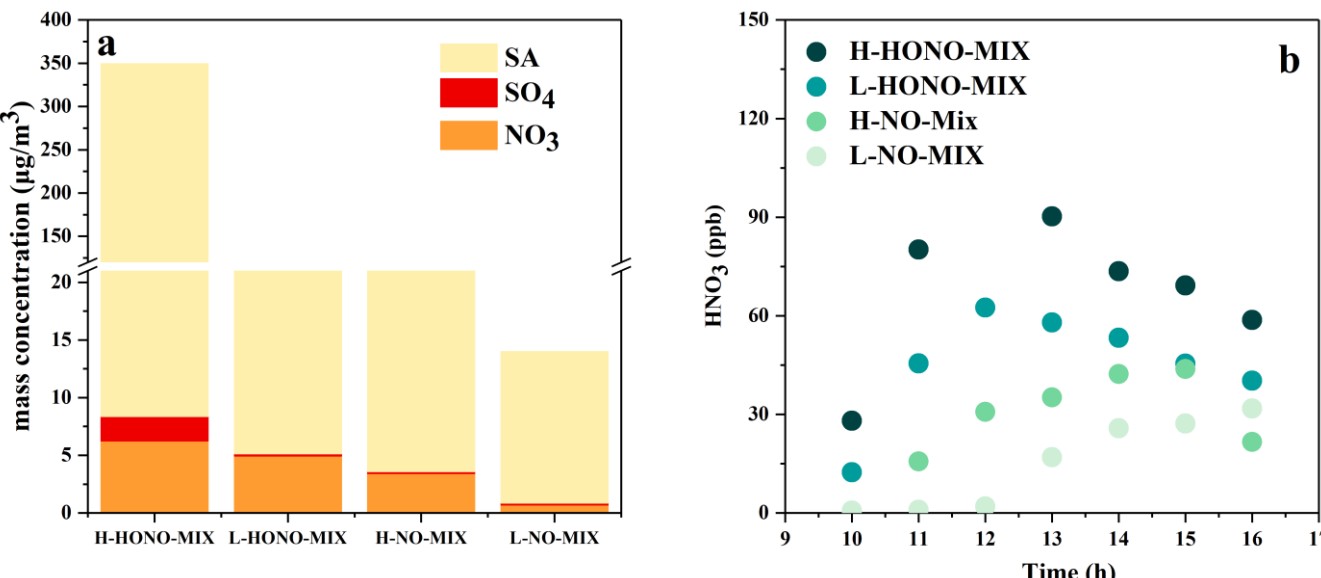

**Figure 6. (a) The measured mass concentration of sulfate, nitrate, and total secondary aerosol in the particle phase. The data used in this plot are the average of the hour in which the mass concentration was the greatest. SA is the abbreviation of secondary aerosol. (b) The concentration of gas-phase nitric acid in the mixture system.**

We analyzed the inorganic components and organic components of the generated particles respectively. Compared with the single-precursor systems, the particle mass concentration generated by the mixture system is the highest, so we selected the mixture system as the target and analyzed the inorganic components. Overall, the amounts of sulfate and nitrate of HONO-Mix experiments are higher than that of NO-Mix experiments (HONO-Mix > NO-Mix). In the above reaction system, the generation pathway of nitric acid is mainly the reaction (Jarvis et al., 2009): $NO_2 + OH \rightarrow HNO_3$. With the similar NO$_x$ concentration, higher OH radicals are beneficial to the generation of nitric acid. As shown in Figure 1, the HONO-Mix

system has higher concentration of OH radicals, which can explain the higher concentration of gas-phase nitric acid (Figure 6b) and nitrate aerosol in this system (HONO-Mix$_{nitrate}$ > NO-Mix$_{nitrate}$). When $SO_2$ is added to the chamber, sulfate is formed by photooxidation of $SO_2$ in the reaction initiated by OH radical (Wang et al., 2021; Liu et al., 2017) (HONO-Mix$_{sulfate}$ > NO-Mix$_{sulfate}$). For HONO experiments, the amounts of sulfate and nitrate of H-HONO-Mix are higher than that of L-HONO-Mix (H-HONO-Mix > L-HONO-Mix), especially for sulfate. As mentioned above, higher OH concentration is

beneficial to nitrate formation, and higher $SO_2$ will promote the formation of sulfate in the system. While for NO experiments, the amounts of sulfate and nitrate of H-NO-Mix are also higher than that of L-NO-Mix (H-NO-Mix > L-NO-Mix), with L-NO-Mix experiment forming negligible sulfate and nitrate. As shown in Figure 1, L-NO-Mix has the lowest OH exposure, and the $SO_2$ concentration is also low, this can explain its low nitrate and sulfate concentration.

As shown in Figure 6a, compared with the total mass concentration of the resulting particles, the sulfate and nitrate

production amounts under the four conditions are very low, so we focus on analyzing the organic components in the particles, including the analysis of functional groups by infrared spectroscopy and the analysis of chemical components by mass spectrometry.

### 3.4.2 Organic chemical components

In order to analyze the functional groups of the particles, we dissolved the collected particles with methanol and detected

them with an infrared spectrometer. Figure S8 and Table S3 show the IR spectra of aerosols formed under NO conditions and HONO conditions. According to the positions of the absorption peak in the IR spectra, different functional groups were assigned. The bold peak at 3360 cm$^{-1}$ is assigned to the characteristic peak of C-OH in alcohol, the broadband at 3100-3300 cm$^{-1}$ (3192 cm$^{-1}$) originates from the O-H stretching vibration of hydroxyl and carboxyl groups (Liu et al., 2017; Coury and Dillner, 2008), and the absorption around 3000-3200 cm$^{-1}$ in 1,3,5-TMB and Mixture experiments can represent the

stretching vibration of C-H bonds in aromatics (Holes et al., 1997). The peaks around 1633-1660 cm$^{-1}$ are assigned to the C=O stretching vibrations of ketones, aldehydes, and carboxylic acids (Coury and Dillner, 2008). The peaks at 2960 cm$^{-1}$ corresponds to $CH_3$ stretching vibration in alkanes, 2921 cm$^{-1}$ and 2850 cm$^{-1}$ corresponds to $CH_2$ stretching vibration in alkanes (Holes et al., 1997), and the broadbands around 1415-1465 cm$^{-1}$ represent the deformation vibrations of methylene and methyl groups. The peak at 1268 cm$^{-1}$ is assigned to the -$ONO_2$ stretching in nitrate ester (Holes et al., 1997; Jia and Xu,

2014; Li et al., 2021b). According to literature reports, peaks in the range of 1000-1200 cm$^{-1}$ is assigned to the absorption peak of sulfate (Wu et al., 2013), peaks around 1040-1070 cm$^{-1}$ represent the absorption band of S=O in organic compound (Chihara, 1958), peak at 1100 cm$^{-1}$ corresponds to the sulfate group in sulfate and organic compounds (Liu et al., 2017). The above analysis confirmed the presence of carboxylic acids, alcohols, nitrates, sulfates, aldehydes, and ketones in aerosols derived from both the NO and HONO conditions with high or low $SO_2$.

In order to conduct a more in-depth analysis of the organic compounds in the particles, we performed the mass spectrometry analysis (Figure 7). In a previous study (Li et al., 2021b), we have shown that the chemical interactions between intermediate products from *n*-dodecane and 1,3,5-TMB can promote particle formation in the mixture experiments

under the HONO conditions. In this study, we focus on the influence of $SO_2$ and the concentration of OH radicals on the formation of particles, and analyzed the formed particles of mixture experiments.

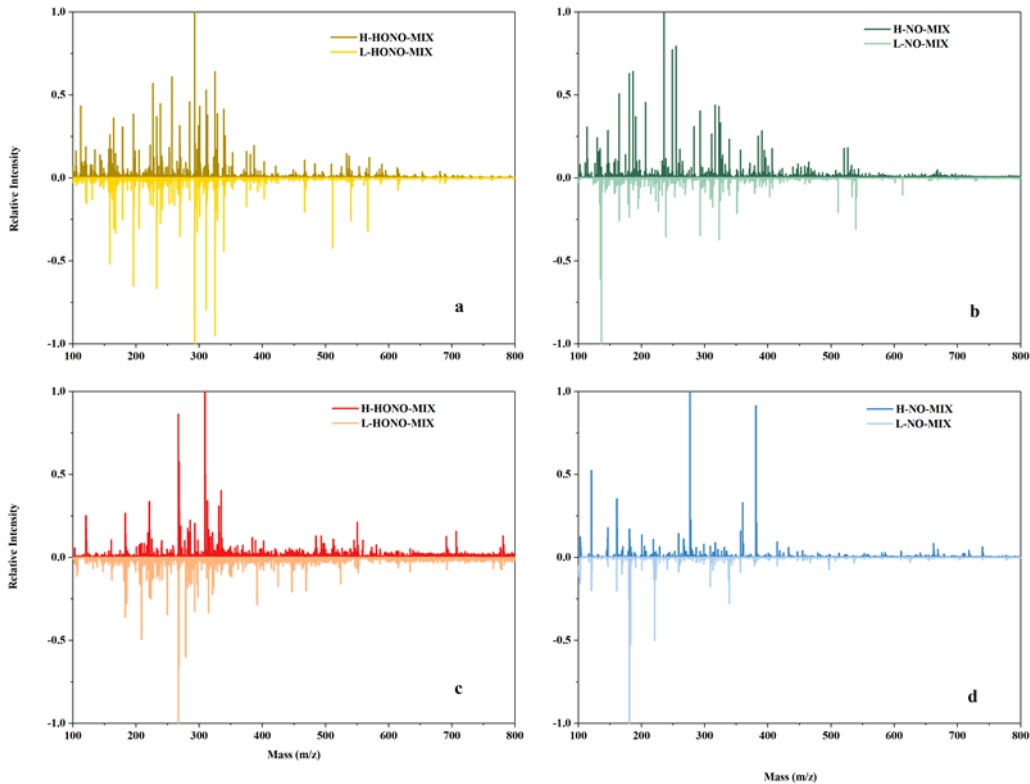


**Figure 7. Mass spectra of mixture experiments (a) HONO mixture experiments in negative mode; (b) NO mixture experiments in negative mode; (c) HONO mixture experiments in positive mode; (d) NO mixture experiments in positive mode. The y axis is the relative intensity normalized by dividing by the maximum signal strength of the mass spectra.**

The products of mixture experiments were detected by ESI-Q-ToF-MS in both negative (Figure 7 a,b) and positive (Figure 7 c,d) mode. Compared to NO-Mix, more products with larger molecular weights are formed under HONO-Mix. This is probably due to the higher OH concentration in the HONO experiments (Figure 1), which favours the formation of large molecular weight products through functionalization reactions. Lambe et al. (2012) reported that when the OH radical exposure was within $(5{\sim}6) \times 10^{11}$ mole cm$^{-3}$ s, the SOA yield of alkanes precursors ($C_{10}$ and $C_{15}$) exhibited an increase as a
function of OH radical exposure, and the increase correlated with an increase in oxygen content.

Meanwhile, organo-sulfates and organo-nitrates are formed in the mixture experiments, as shown in Table S4. It is found that larger concentration organo-nitrates are formed under HONO conditions compared with NO conditions. The primary formation pathway of organo-nitrates is the reaction of $RO_2 + NO$, and the $RO_2$ is mainly formed through the reaction of organic gases with OH radical (Li et al., 2022; Tsiligiannis et al., 2019). Under HONO conditions, higher
concentration of OH radical is formed (Figure 1), so more $RO_2$ will exist in HONO experiments, and thus more organo-

nitrates will be formed. In addition, high $SO_2$ conditions are more conducive to the formation of organo-sulfates, e.g., H-NO-$Mix_{organo-sulfate}$ > L-NO-$Mix_{organo-sulfate}$ (Table S4). As discussed in section 3.4.1, sulfate is formed by reaction of $SO_2$ + OH radical (Wang et al., 2021; Liu et al., 2017), and higher $SO_2$ would facilitate the formation of sulfate in the experiments. According to previous studies (Yang et al., 2020), organo-sufates are mainly formed through the reaction of sulfate with

compound containing OH bonds or ether bonds. Therefore, higher $SO_2$ would facilitate the formation of organo-surfates.

The proposed reaction mechanism of mixture experiment in the presence of $NO_x$ and $SO_2$ is shown in Figure 8. The products derived from *n*-dodecane are oxygen-containing organic compounds (aldehydes, ketones, alcohols, carboxylic acid, etc.), organic nitrates, and organo-surfates; the products derived from 1,3,5-TMB are also multifunctional products containing carbonyl, acid, alcohol, nitrate, and sulfate functional groups. For the mixture experiment, the intermediate

multifunctional products can react with each other, high molecular weight oligomers are thus produced. As an example, $C_{14}H_{21}NO_5$ might be the product from the reaction of a phenol from 1,3,5-TMB ($C_9H_{12}O$) and an aldehydes from *n*-dodecane ($C_5H_{10}O$).

Figure 8. Proposed reaction mechanism of the mixture experiment in the presence of NOx and SO₂ (R1 and R2 are alkyl groups). Blue texted compounds are detected by ESI-Q-ToF-MS in this work; solid boxed compounds are detected by previous studies (black, Yang et al., 2020; purple, Huang et al., 2014; green, Sato et al., 2019; orange, Yee et al., 2013). The reactions in the dotted boxes are the proposed reaction paths of the mixture experiment.

## 4 Conclusion and implications

In the present work, a large-scale outdoor smog chamber was applied to study the effect of inorganic gases ($SO_2$ and $NO_x$) on the photochemical process of mixed anthropogenic organic gases, i.e., *n*-dodecane and 1,3,5-trimethylbenzene. The OH concentration under the HONO conditions is higher than that under the classic $NO_x$ conditions at similar measured $NO_x$ concentration. The ozone formation is affected by the reaction precursors, the concentration of OH radicals, and the

temperature: precursors with higher ozone formation potential also contribute significantly to ozone formation in the mixture reaction system; higher temperature and higher OH concentration are beneficial to the formation of ozone in the reactions. In contrast, the presence of $SO_2$ has little effect on the concentration of ozone.

However, the presence of $SO_2$ can greatly promote the formation of particles in both number and mass concentration, likely due to the enhanced new particle formation and acid-catalyzed heterogeneous reactions from the formation of sulfuric acid and the formation of organo-sulfates (Liu et al., 2019; Liu et al., 2017; Li et al., 2017c). In addition, higher OH radical concentration and lower temperature are also beneficial to the formation of particles. For the particle composition, the content of inorganic nitrate and sulfate under the HONO conditions was higher than that under the NO conditions, although organic aerosols dominate the total secondary aerosols. The organo-sulfates and organo-nitrates are detected in the formed particles, and the presence of $SO_2$ is found to promote the formation of organo-sulfates.

This study provides the first attempt to investigate the role of $SO_2$ in the oxidation of mixed anthropogenic organic gases with various OH concentrations and temperature conditions. The results here can improve our understanding in the chemical processes that lead to ozone and secondary particle formation in the complex urban areas influenced by complex emissions (including vehicle exhaust, coal combustion, etc.). More in-depth and detailed research on the mixture reaction systems with atmospheric-relevant conditions should be carried out in the future to deepen our understanding of the physical and chemical processes in the oxidation of organic vapors in the real atmosphere.

*Data availability.* The data used in this study are available upon request from the corresponding author.

*Author contributions.* JLL, HL, and MFG designed the experiments; JLL conducted the experiments with help from HZ, XZ, YYJ, WHC and YXK; JLL analyzed the data and wrote the paper, with contributions from KL, HL and MFG; and YXC, YQR, YJZ, HJZ, RG, ZHW, FB, XC, XZW, and WGW commented on the paper.

*Competing interests.* The authors declare that they have no conflict of interest

*Acknowledgements.* We are grateful to Professor Likun Xue and postgraduate Xuelian Zhong from Shandong University for providing the MCM model.

*Financial support.* This research has been supported by the National Natural Science Foundation of China (Contract No. 42130606), Special fund project of Guangdong Provincial Department of Ecology and Environment, the Beijing Municipal Science & Technology Commission (Grant no. Z181100005418015), National research program for Key issues in air pollution control (DQGG2021301), and the Fundamental Research Funds for Central Public Welfare Scientific Research Institutes of China, Chinese Research Academy of Environmental Sciences (No. 2019YSKY-018).

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
