# Peer review of "Effects of OH radical and SO2 concentrations on photochemical reactions of mixed anthropogenic organic gases"

_Atmospheric Chemistry and Physics, 2022_

## Author Response (AR1)

**Response to the comments of Reviewer #1**

*The manuscript entitled "Effects of OH radical and SO$_2$ concentrations on photochemical reactions of mixed anthropogenic organic gases" presents new findings of the atmospheric processing of the anthropogenic pollutants, represented by n-dodecane, 1,3,5-trimethylbeneze, which are associated with the vehicular traffic pollution. The paper is scientifically sound; for the most part, methods and experimental details are adequately presented. The equipment and methodology used in the simulation chamber experiments are adequate and provide valuable information about the reactions under investigation. The length of sections 1 and 2 is well balanced, providing sufficient details and discussion without adding too much volume to the final manuscript.*

Response: We thank Anonymous Referee #1 for the review and the positive evaluation of our manuscript. We have fully considered the comments and responded to these comments below in blue text. The revisions in the manuscript are highlighted in yellow color. The response and changes are listed below.

*Major Comments*

1. *At the same time, the article would benefit from major revisions. Generally, the use of the English language should be improved because it is frequently awkward, even from the point of view of a non-native speaker of English.*

    We thank the reviewer for pointing this out. The use of the English language has been polished.

2. *My technical comments are provided below. Regarding the scientific comments, I believe that the article would benefit greatly from a more in-depth analysis of the results. The is a lack of a broader context in the presented discussion. The discussion in section 3 discusses the data but without providing any broader insights into the processes under investigation. In connection with this comment, there is no quantitative information presented in section 4, which almost reads like a literature review section. I would recommend constructing a kinetic model (perhaps MCM can be utilized in some way) and attempting to reproduce the experimental temporal profiles of the reactants from the chamber experiments and the measured yields of SOAs. All of the elements are here; rate coefficients for the two molecules under investigation are available or can be estimated with SAR parameter if needed. The author should attempt to construct a mechanism explaining the*

*experimental observations and the use of this mechanism to discuss and explain the atmospheric*

*implications of their findings in section 4 (Atmospheric Implications).*

*See also:*

*Environ. Sci. Technol. 2001, 35, 1394-1405*

*https://acp.copernicus.org/articles/22/215/2022/acp-22-215-2022.pdf*

We thank the reviewer for this comment. We agree that a kinetic model (e.g., MCM) reproducing the time series could help to improve this paper. However, modeling SOA formation is very complex, which needs not only the reaction mechanism in the gas phase, but also complex gas–particle partitioning process which is related to volatility, phase, and mixing state. Therefore, modeling SOA formation, in our opinion, is greatly beyond the scope of this study, and might be a separate paper itself. Nonetheless, we have applied the MCM model to simulate the experimental temporal profiles of the reactants and ozone, and a mixture experiment was selected for comparison with the simulated results. (Page 9-10, line 228-238, Figure 3)

"Figure 3 shows the concentration-time profiles of measured and simulated ozone and ozone formation and loss rates in H-HONO-Mix-4 experiment. The experiment was simulated with Master Chemical Mechanism MCM version3.3 (http://mcm.leeds.ac.uk/MCM/). The model was constrained with measured NO, $NO_2$, and HONO concentration. As shown in Figure 3a, the ozone production is well represented by the model in the first 0.5 h, however the model starts to over-predict the $O_3$ concentration in the after 5.5 h. This phenomenon is similar to a study about 1,3,5-TMB, the experiment of which was performed with an outdoor chamber (Metzger et al., 2008). Meanwhile, the ozone photochemical budget cycle was simulated with MCM model. The two major reactions of $HO_2$ + NO and $RO_2$ + NO control the photochemical generation of ozone; $NO_2$+OH, $NO_2$+ $RO_2$, and VOCs + $NO_3$ reactions control the ozone consumption."

[Figure]

Figure 3. Concentration-time profiles of monitored and simulated (a) ozone, NO, $NO_2$, and 1,3,5-TMB and dodecane, and (b) ozone formation and loss rates in the H-HONO-Mix-4 experiment.

According to literature and experiment results, the reaction mechanism is proposed, and it has been added in the manuscript. (Page 15-16, line 356-367, Figure 8).

[Figure]

Figure 8. Proposed reaction mechanism of the mixture experiment in the presence of NOx and SO$_2$ (R1 and R2 are alkyl groups). Blue texted compounds are detected by ESI-Q-ToF-MS in this work; solid boxed compounds are detected by previous studies (black, Yang et al., 2020; purple, Huang et al., 2014; green, Sato et al., 2019; orange, Yee et al., 2013). The reactions in the dotted boxes are the proposed reaction paths of the mixture experiment.

Yang, Z., Tsona, N. T., Li, J., Wang, S., Xu, L., You, B., and Du, L.: Effects of NOx and SO$_2$ on the secondary organic aerosol formation from the photooxidation of 1,3,5-trimethylbenzene: A new source of organosulfates, Environmental pollution, 264, 114742, 10.1016/j.envpol.2020.114742, 2020.

Huang, M., Hu, C., Guo, X., Gu, X., Zhao, W., Wang, Z., Fang, L., and Zhang, W.: Chemical composition of gas and particle–phase products of OH–initiated oxidation of 1,3,5–trimethylbenzene, Atmos. Pollut. Res., 5, 73-78, 10.5094/apr.2014.009, 2014.

Sato, K., Fujitani, Y., Inomata, S., Morino, Y., Tanabe, K., Hikida, T., Shimono, A., Takami, A., Fushimi, A., Kondo, Y., Imamura, T., Tanimoto, H., and Sugata, S.: A study of volatility by composition, heating, and dilution measurements of secondary organic aerosol from 1,3,5-trimethylbenzene, Atmos. Chem. Phys., 19, 14901-14915, 10.5194/acp-19-14901-2019, 2019.

Yee, L. D., Craven, J. S., Loza, C. L., Schilling, K. A., Ng, N. L., Canagaratna, M. R., Ziemann, P. J., Flagan, R. C., and Seinfeld, J. H.: Effect of chemical structure on secondary organic aerosol formation from C12 alkanes, Atmos. Chem. Phys., 13, 11121-11140, 10.5194/acp-13-11121-2013, 2013.

_Technical comments._

3.  _Line 36-37, 41-45 These sentences are not well constructed and read awkwardly, please revise._

    Thank you for this comment. These sentences have been revised. (Page 2, line 36-38, line 42-46)

4.  _etc. is used a little bit too much in the introduction, please avoid such abbreviations in the scientific writing._

    This has been revised in the introduction.

5.  _Line 61 Consider removing "in combination with the corresponding equipment"_

    This has been removed.

6.  _Lines 74-77 Can you please clarify why the temperature inside the chamber during wintertime is within 15-30 °C range?_

    As the bottom of the reactor was made of aluminum, after a period of sunlight exposure, the surface temperature of the aluminum plate will rise. The chamber covered by Teflon film is equivalent to a greenhouse, the internal temperature will rise after the sunlight exposure. The cooling system of the chamber is water-cooled, in order to prevent the cooling pipes from being frozen and cracked, the system is closed. Thus, the temperature inside the chamber during winter is higher than the ambient environment. The corresponding explanation has been added in the manuscript. (Page 3, line 82-86)

7.  _Line 80 Consider removing "classics"_

    This has been removed.

8.  _Line 80 NO was introduced from a 500 ppm standard gas cylinder, I understand that this cylinder contained a 500 ppm mixture of NO in nitrogen? Similar comment to Line 84 (SO₂ cylinder)._

    Yes, NO cylinder contained a 500 ppm mixture of NO in nitrogen, and $SO_2$ cylinder contained a 60 ppm mixture of $SO_2$ in nitrogen. The corresponding text has been added in the manuscript. (Page 3, line 93,96)

9.  _Line 82 Consider removing "home-made"_

    This has been removed.

10. *Line 92 Consider removing "solid" and changing adsorbent to sorbent*

   "Solid" has been removed and "adsorbent" has been changed to "sorbent". (Page 4, line 105)

11. *Line 99 Can you provide some more details about the experimental conditions for the ESI-MS measurements? Perhaps in the SI? What was the difference between the measured and expected elemental formula? What was the mass resolution of the used instrument? Note also that the elemental composition provides little information about the molecular structure.*

   The experimental conditions for the ESI-MS measurements have been added in the "2.2 online and offline measurement part" and Supporting Information. (Page 4, line 113-116; Supporting information, Page 1, line 10-17)

   In general, the theoretical molecular mass of a compound refers to the monoisotopic mass of the compound, which is the sum of the masses of its constituent elements, with each element mass choosing the mass of its most abundant isotope. The expected elemental formula means the formula with the theoretical molecular mass. However, due to the error/interference of the instrument, the measured molecular mass is usually slightly different from the expected molecular mass, and this difference is indicated as mass accuracy (in ppm). The mass resolution of this used instrument is $>$ 20000. In order to explain the detected compounds in depth, we have compared the detected substances with the existing literatures, and this has been added in the manuscript.

12. *Line 111 Referring to OH as the hydroxyl free radical is rather uncommon.*

   This has been revised in the manuscript.

13. *Lines 271-272 more as a higher number or larger concentration?*

   "More" has been changed to "larger concentration". (Page 14, line 347)

14. *Figures S2 and S3 – the labels on these plots are completely unreadable, can you please make the fonts larger?*

   The labels on Figure S2 and S3 have been revised. (Supporting information, Page 7-9, line 51-67)

15. *Figures 2 and 4 are difficult to read, perhaps consider presenting some of these results in a form of a bar plot?*

   Thank you for your comment. Figure 2 and 4 have been revised, now as Figure 2 and Figure 5. The color-coding of shaded areas in the plot have been removed. In addition, the corresponding traditional plots (one variable vs. one factor) have been added in the Supporting Information as Figure S4 and S7. (Supporting Information, Page 9-10, line 68-70; Page 12, line 85-87)

[Figure]

Figure S4. Ozone formation in the NO and HONO experiments. The temperature (T) and ozone concentration here refers to the maximum value during the reaction process.

[Figure]

Figure S7. Particle formation of NO and HONO experiments. The temperature (T) and particle mass concentration here refers to the maximum value during the reaction process.

**Response to the comments of Reviewer #2**

*The manuscript presents smog-chamber studies of the photooxidation of n-dodecane, 1,3,5-trimethylbeneze, and their mixture in the presence of OH, NOx, HONO, and SO$_2$. The purpose was to simulate the photooxidation of vehicle exhausts in urban atmospheres. The experiments were carried out in an outdoor chamber illuminated by sunlight. The authors followed NOx, SO$_2$, O$_3$, HONO, and HNO$_3$ concentrations in the chamber and precursor concentrations before and after the reactions. They monitored the number and mass of particles formed and collected them on PTFE filters for subsequent direct-injection ESI-MS and FTIR analyses. Besides, they analyzed inorganic nitrite and sulfate contents in the particles. The results showed that ozone formation during the reactions was enhanced by OH radicals and temperature but not by SO$_2$. On the other hand, SO$_2$ increased the number and mass of particles formed. The particles contained many organic compounds, including organosulfates and organonitrates.*

*The transformation of vehicle exhaust in urban atmospheres, including particulate matter formation and its composition, is a relevant topic of atmospheric chemistry and air quality studies. The authors present and analyze new experimental data on the photooxidation of two components of the exhausts and their mixtures. The presented results and analysis should be interesting for ACP readers and deserve publication in that journal, provided the submitted manuscript is corrected and extended. Below is a list of necessary corrections and extensions before ACP editors may accept the manuscript.*

Response: We thank Anonymous Referee #2 for the review and the positive evaluation of our manuscript. We have fully considered the comments and responded to these comments below in blue text. The revisions in the manuscript are highlighted in yellow color. The response and changes are listed below.

**Introduction**

1. *The author might cite a recent review (Srivastava et al. 2022) to support the relevance of their work.*
   This review has been added in the introduction part of the manuscript. (Page 2, line 35)

   Srivastava, D., Vu, T. V., Tong, S., Shi, Z., and Harrison, R. M.: Formation of secondary organic aerosols from anthropogenic precursors in laboratory studies, npj Climate and Atmospheric Science, 5, 10.1038/s41612-022-00238-6, 2022.

2. *The authors name several groups of compounds investigated by researchers (e.g., long-chain*

*alkanes and aromatic hydrocarbons). They may consider explicitly naming a few examples that were*

*studied in the cited works.*

Thank you for your suggestion. The corresponding examples have been added in the manuscript. (Page 2, line 38-39; line 47)

3. *Lines 59-63. The authors should justify the choice of n-dodecane (DOD) and 1,3,5-trimethylbeneze (TMB) as the model compounds for the study. Besides, they should briefly explain the purpose of comparing NO and HONO experiments (also applies to Section 2.1, lines 73-82).*

According to the field observation in China, high concentration of 1,3,5-TMB and *n*-dodecane were observed: the 1,3,5-TMB concentration at rural site could reach 1.447 ppb, and the measured concentration of $C_{12}$ alkanes at rural site was $0.122\pm0.12$ ppb (Chen et al., 2020; Wang et al., 2020a). In addition, 1,3,5-TMB and *n*-dodecane are important components in liquid gasoline and diesel and their vapors (Schauer et al., 2002; Gentner et al., 2012). This has been added in the manuscript. (Page 2, line 61-64)

NO and HONO experiments were designed, as their pathways for generating OH radicals were different. In HONO experiments, the photolysis of HONO could directly generate OH radicals; while for NO experiments, the generation of OH radical was through recycling via $NO_x/HO_x$ chemistry (Ng et al., 2007). Therefore, the OH concentration and exposure are also different in these two experimental setups. This has been added in the manuscript. (Page 3, line 89-92)

Chen, T., Xue, L., Zheng, P., Zhang, Y., Liu, Y., Sun, J., Han, G., Li, H., Zhang, X., Li, Y., Li, H., Dong, C., Xu, F., Zhang, Q., and Wang, W.: Volatile organic compounds and ozone air pollution in an oil production region in northern China, Atmospheric Chemistry and Physics, 20, 7069-7086, 10.5194/acp-20-7069-2020, 2020.

Gentner, D. R., Isaacman, G., Worton, D. R., Chan, A. W. H., Dallmann, T. R., Davis, L., Liu, S., Day, D. A., Russell, L. M., Wilson, K. R., Weber, R., Guha, A., Harley, R. A., and Goldstein, A. H.: Elucidating secondary organic aerosol from diesel and gasoline vehicles through detailed characterization of organic carbon emissions, Proceedings of the National Academy of Sciences of the United States of America, 109, 18318-18323, 10.1073/pnas.1212272109, 2012.

Ng, N. L., Kroll, J. H., Chan, A. W. H., Chhabra, P. S., Flagan, R. C., and Seinfeld, J. H.: Secondary organic aerosol formation from m-xylene, toluene, and benzene, Atmospheric Chemistry and Physics, 7, 3909-3922, 2007.

Schauer, J. J., Kleeman, M. J., Cass, G. R., and Simoneit, B. R. T.: Measurement of emissions from air pollution sources. 5. C-1-C-32 organic compounds from gasoline-powered motor vehicles, Environmental Science & Technology, 36, 1169-1180, 10.1021/es0108077, 2002.

Wang, C., Yuan, B., Wu, C., Wang, S., Qi, J., Wang, B., Wang, Z., Hu, W., Chen, W., Ye, C., Wang, W., Sun, Y., Wang, C., Huang, S., Song, W., Wang, X., Yang, S., Zhang, S., Xu, W., Ma, N., Zhang, Z., Jiang, B., Su, H., Cheng, Y., Wang, X., and Shao, M.: Measurements of higher alkanes using NO+ chemical ionization in PTR-ToF-MS: important contributions of higher alkanes to secondary organic aerosols in China, Atmospheric Chemistry and Physics, 20, 14123-14138, 10.5194/acp-20-14123-2020, 2020.

4. *Line 44. What is "S/IVOCs"?*

   S/IVOCs are the abbreviations of semi/intermediate volatile organic compounds. This has been added in the manuscript. (Page 2, line 44-45)

*Section 2.1. Smog chamber experimental conditions.*

5. *Line 74. Was there any particular reason for carrying out the experiments only in winter?*

   Temperature of the experiments carried out in winter in this chamber was in the range of 15~30 ℃,and this was suitable for studying reactions at room temperature.

*Section 2.2. Online and offline measurements*

6. *Lines 92-95. The concentrations of organic precursors determined before and after the photooxidation do not appear in the manuscript. Tables S1 and S2 show the initial concentrations, but it is not clear if they were measured or calculated.*

   The concentration of organic precursors was measured before and after the photooxidation. As the concentration of organic precursors after the photooxidation was nearly zero, so the initial concentrations were shown in Table S1 and S2 to represents the organic precursors consumed in the reaction. This has been added in the manuscript. (Page 4, line 107-109)

7. *Lines 98-99. The authors should describe how the filter extraction was done exactly (whole filter or punches, volume of methanol, time of extraction, device used for extraction).*

   The collected whole PTFE filter was extracted with 5 mL methanol in an ultrasonic bath (KH5200DV, Hechuang Ultrasonic, China) for 30 min. This has been added in the manuscript. (Page 4, line 113-114)

8. *Lines 99-101. The authors should specify the inorganic species they analyzed in the gas and particle phases.*

   The measured inorganic species in gases was nitric acid, and the inorganic species measured in aerosols included sulfate and nitrate. This has been added in the manuscript. (Page 4, line 118-119)

*Section 2.3. Calculation methods of SOA yields and OH concentration*

9. *Lines 105-110. The authors should provide more details on the wall-losses analysis. Mere reference to another paper may be insufficient for readers.*

   Details of the wall-loss analysis have been added in the manuscript. (Page 4-5, line 127-152)

10. *Lines 111-121. The calculation of the OH concentration was essential for the analysis of the results presented, so it should be described in better detail. Namely:*

*The condition of constant OH concentration and integration of Eqn (1) are not necessary since the slope of the logarithmic TMB time profile is always equal to k[OH]*

$$d \ln ([TMB])/dt = -k [OH] \qquad (R\text{-}1)$$

The calculation of the OH concentration was described in details, and the added and revised parts have been colored in yellow in the manuscript. (Page 5-6, line 153-165)

11. *Line 120. The authors should specify how they averaged the slope and [OH].*

    As mentioned in question 10, this part has been added in the method part. (Page 5-6, line 153-165)

12. *In Section 2.2, the authors should describe how TMB concentrations were determined and with what time resolution for [OH] calculations. Besides, how was [OH] determined in n-dodecane experiments in which TMB was absent?*

    In this work, 1,3,5-trimethylbenzene is chosen as the tracer for the mixture experiments, as it reacts mainly with OH and has no interference with other compounds. Combined with the sampling frequency, the time resolution for [OH] calculation is about 1-1.5 hour. The [OH] concentration is only calculated and compared in the mixture experiments. The corresponding content has been added in the manuscript. (Page 6, line 166-169)

*Section 3.1. General results of the experiments*

13. *Lines 130 and 140 (Figure 1 caption). The authors should explain the term "exposure" and how they calculated it.*

    The explanation of OH exposure has been added in Section 2.3. (Page 6, line 166-169)

14. *The manuscript should include time profiles of the organic precursors studied (at least in Figures S2 and S3), which would help readers understand the reaction progress.*

    The time profiles of organic precursors have been added in Figures S2 and S3. (Supporting Information, Page 7-9)

15. *In line 140, Figure 1 should show when the chamber enclosure was opened precisely. Line 76 says the enclosure was opened between 9:00 and 10:00 a.m., but the first experimental points in Figure 1 are at 11:00 a.m. If each point is some average over the time window, that window should also be specified.*

    The opening time of the chamber enclosure is the time when the photochemical reaction begins, while the points in Figure 1 means the average OH concentration (Figure 1a) and OH exposure (Figure 1b) in the time interval between this data point and the previous data point. For the first

experimental points in Figure 1, they are the average OH concentration (Figure 1a) and OH exposure (Figure 1b) in the time interval between 11:00 a.m. and the opening time of the photochemical reactions. The explanation has been added in the manuscript. (Page 7, line 188-191)

*Section 3.2. Ozone formation and gas phases products*

16. *Line 150. Figure 2 includes several encodings referring to precursors, reactants, and reaction parameters. A reader needs some effort to decipher those encodings, so it would be beneficial to have them explicitly explained in the Figure caption. Besides, the color-coding of shaded areas in the plot seems redundant with other encodings, so that it could be removed for presentation simplicity.* The corresponding explanation has been added in the Figure caption, and the color-coding of shaded areas in the plot have been removed. (Page 8, line 199-207)

[Figure]

Figure 2. Ozone formation in the NO and HONO experiments. The temperature (T) and ozone concentration here refers to the maximum value during the reaction process. The yellow circles refer to NO-Dod experiments with low $SO_2$ concentration, the yellow boxes refer to NO-Dod experiments with high $SO_2$ concentration; the yellow filled circles refer to HONO-Dod experiments with low $SO_2$ concentration, and the yellow filled boxes refer to HONO-Dod experiments with high $SO_2$ concentration. The blue circles refer to NO-TMB experiments with low $SO_2$ concentration, the blue boxes refer to NO-TMB experiments with high $SO_2$ concentration; the blue filled circles refer to HONO-TMB experiments with low $SO_2$ concentration, and the blue filled boxes refer to HONO-

TMB experiments with high $SO_2$ concentration. The red circles refer to NO-Mix experiments with low $SO_2$ concentration, the red boxes refer to NO-Mix experiments with high $SO_2$ concentration; the red filled circles refer to HONO-Mix experiments with low $SO_2$ concentration, and the red filled boxes refer to HONO-Mix experiments with high $SO_2$ concentration.

17. *Line 178. Write explicitly "concentrations similar to those in NO experiments."*

This has been revised in the manuscript. (Page 10, line 247-248).

*Section 3.3. Effect of $NO_x$ and $SO_2$ on particle formation*

18. *Line 188. Figures S4 and S5 show that in some cases, a banana-like formation of particles occurred, while in other cases not. Could the authors discuss that observation briefly?*

When the max number concentrations are above $10^5$ #/cm$^3$, the banana like-formation particles occur, especially in experiments with high $SO_2$ concentration. The corresponding explanation has been done in the manuscript. (Page 10, line 256-258)

19. *Line 200. The OH concentration in HONO experiments was higher than in NO experiments only in the initial hours.*

This expression has been revised in the manuscript. This phenomenon can be explained by higher OH exposure in HONO experiments, as shown in Figure 1b. Higher OH exposure causes the higher consumption rate of the precursors and the subsequent faster particle generation rates. (Page 11, line 267-270)

20. *Lines 202-205 and 210 (Figure 4). I like the encoding concept of Figures 2 and 4 but inferring some relations from them is not easy. For instance, the influence of temperature on particle mass mentioned in line 204 seems not monotonous. For such comparisons, I would like to see traditional plots (one variable vs. one factor) in the Supplementary Information.*

For Figure 2 and 4, the corresponding traditional plots have been added in the Supporting Information, as shown in Figure S4 and S7. (Supporting Information, Page 9-10, line 68-70; Page 12, line 85-87)

[Figure]

Figure S4. Ozone formation in the NO and HONO experiments. The temperature (T) and ozone concentration here refers to the maximum value during the reaction process.

[Figure]

Figure S7. Particle formation of NO and HONO experiments. The temperature (T) and particle mass concentration here refers to the maximum value during the reaction process.

21. *Line 210 (Figure 4). I have comments on encoding, same as for Figure 2 (Line 150).*

The explanations of the encodings have been added in the Figure caption, the color-coding of shaded areas in the plot have been removed. (Page 11-12, line 277-286)

*Section 3.4. Chemical composition of particles*

22. *Line 217 (Figure 5). In the caption, mark what SA is. The sulfate and nitrate bars would be more visible if the mass axis in panel (a) was broken, say between 20 and 90 or 90 and 300.*

Figure 5 (now is Figure 6) has been revised according to your suggestion. (Page 12, line 289-292)

[Figure]

Figure 6. (a) The measured mass concentration of sulfate, nitrate, and total secondary aerosol (SA) in the particle phase. The data used in this plot are the average of the hour in which the mass concentration was the greatest. (b) The concentration of gas-phase nitric acid in the mixture system.

23. *Line 240. Correct to "Organic chemical composition."*

This has been corrected in the manuscript. (Page 13, line 313)

24. *Lines 240-254. The authors might compare the FTIR observations with literature e.g., (Holes et al. 1997).*

Thank you for this suggestion, the FTIR observations have been compared with literature, and corresponding part has been added in the manuscript. (Page 13, line 319-324)

Holes, A., Eusebi, A., Grosjean, D., and Allen, D. T.: FTIR analysis of aerosol formed in the photooxidation of 1,3,5-trimethylbenzene, Aerosol Sci. Technol., 26, 516-526, 10.1080/02786829708965450, 1997.

25. *Lines 255-280. Proper mass spectrometric analysis of particulate matter should include the separation of analytes, e.g., by chromatography. The direct-injection method used by the authors is less informative, challenging to interpret, and may serve only as "a first glance approach." The authors might compare the list of ions observed (Table S-3) with literature, e.g. (Praplan et al. 2014; Sato et al. 2012).*

Thank you for this comment. We compare the list of ions observed with literature, and the proposed reaction mechanism has been added in the manuscript. (Page 15-16, line 356-367)

Figure 8. Proposed reaction mechanism of the mixture experiment in the presence of NOx and SO$_2$ (R1 and R2 are alkyl groups). Blue texted compounds are detected by ESI-Q-ToF-MS in this work; solid boxed compounds are detected by previous studies (black, Yang et al., 2020; purple, Huang et al., 2014; green, Sato et al., 2019; orange, Yee et al., 2013). The reactions in the dotted boxes are the proposed reaction paths of the mixture experiment.

Yang, Z., Tsona, N. T., Li, J., Wang, S., Xu, L., You, B., and Du, L.: Effects of NOx and SO2 on the secondary organic aerosol formation from the photooxidation of 1,3,5-trimethylbenzene: A new source of organosulfates, Environmental pollution, 264, 114742, 10.1016/j.envpol.2020.114742, 2020.

Huang, M., Hu, C., Guo, X., Gu, X., Zhao, W., Wang, Z., Fang, L., and Zhang, W.: Chemical composition of gas and particle–phase products of OH–initiated oxidation of 1,3,5–trimethylbenzene, Atmos. Pollut. Res., 5, 73-78, 10.5094/apr.2014.009, 2014.

Sato, K., Fujitani, Y., Inomata, S., Morino, Y., Tanabe, K., Hikida, T., Shimono, A., Takami, A., Fushimi, A., Kondo, Y., Imamura, T., Tanimoto, H., and Sugata, S.: A study of volatility by composition, heating, and dilution measurements of secondary organic aerosol from 1,3,5-trimethylbenzene, Atmos. Chem. Phys., 19, 14901-14915, 10.5194/acp-19-14901-2019, 2019.

Yee, L. D., Craven, J. S., Loza, C. L., Schilling, K. A., Ng, N. L., Canagaratna, M. R., Ziemann, P. J., Flagan, R. C., and Seinfeld, J. H.: Effect of chemical structure on secondary organic aerosol formation from C12 alkanes, Atmos. Chem. Phys., 13, 11121-11140, 10.5194/acp-13-11121-2013, 2013.

*References*

26. *Line 501. Correct "Tadeusz E. Kleindienst" to "Kleindienst, Tadeusz E." and move the reference to the correct place on the list.*

   This reference has been revised and moved to the correct place on the list. (Page 21, line 509-511)

*Supporting information*

27. *Tables S1 and S2. Explain "Mo" in table headings.*

   "Mo" has been changed to "M", which is the mass concentration of formed secondary aerosol. The explanation has been added in table headings. (Page 2-3, line 29-35)

28. *Table S3. Explain "RDB" in table heading.*

   Now Table S3 is Table S4, RDB is the abbreviation of ring and double-bond equivalent, this has been added in table heading. (Page 5, line 41)

29. *Figures S2 and S3 are illegible even after magnification in the pdf file and must be improved.*

   Figure S2 and S3 have been improved. (Page 6-9)

---

## Editor Decision (ED1)

**Additional comments on acp-2022-294**

Based on the comments of two experts in the field, and after my consideration, the manuscript is of adequate atmospheric interest to merit publication in Atmospheric Chemistry and Physics. The authors have thoroughly responded all the questions/comments raised by the reviewers, and modified the manuscript according to the suggestions.

However, I have some additional comments, which are needed to be solved before publication.

1. Please, update also the last section with atmospheric implications (see below)
2. The statement "The data used in this study are available upon request from the corresponding author" needs to be changed. Namely, data should be available in the repositories. Please, check the manuscript preparation guidelines for authors. https://www.atmospheric-chemistry-and-physics.net/policies/data_policy.html.

**Technical comments/ errors:**
Line 36: Still not ok. Correct as: »Laboratory studies of long-chain alkanes, as representatives of IVOCs, are mainly focused on the case of a single long-chain alkanes or mixture of various precursors, which include long….

Line 42: The sentence needs to be corrected in a way: "For the mixture of various precursors including long-chain alkanes, studies are mainly focused on the chemical composition of the mixture gases, the properties of total organic carbon, the amount of SOA generated, …

Line 86:…than in the ambient environment
Line 90: Correct as:"… as the OH precursor; therefore, NO and HONO experiments were designed, as their pathways for generating OH radicals…."

Line 102: "Gaseous NOx, SO2, and O3 concentrations inside the chamber were monitored in real time by an SO2 analyzer…« Please change as: »…by NOx (),…SO2 () and O3 () analyzers.«

Line 118/119: The sentence can be deleted; but "sulfate and nitrate" can be added in the parentheses after "in aerosols ()" and "nitric acid" after gases ().

Lines 188-191: Can you write more understandable?

Line 232: Please correct: »to over predict the O3 concentration in the after 5.5 h«. (after 0.5 or 5.5h?)

Line 233: Correct the sentence: "This phenomenon is similar to a study about 1,3,5-TMB, the experiment of which was performed with an outdoor chamber"

Line 313: Correct to "Organic chemical composition"
Line 347: ….larger concentration of organo-nitrates…

Line 358: Instead of organo-surfates should be organo-sulfates"

Please, revise/edit:

**4. Conclusions and atmospheric implications**:

Please revise/edit the last section with atmospheric implications. According to the suggestion of Reviewer 1, you constructed a mechanism explaining the experimental observations; but try (if possible) to use this mechanism in discussion and explanation of the atmospheric implications of your findings.

---

## Author Response (AR3)

Dear Editor,

Thank you very much for handling our manuscript submitted to *Atmospheric Chemistry and Physics* (MS No.: acp-2022-294; Title: Effects of OH radical and SO2 concentrations on photochemical reactions of mixed anthropogenic organic gases).

We have addressed all your comments and revised our manuscript very carefully. To proceed, we have uploaded three files, including 1) our point-to-point reply; 2) the revised manuscript with changes highlighted in yellow; 3) the revised manuscript without track-changes.

On behalf of all the co-authors, I would like to thank you and referees for all the invaluable comments. Please feel free to contact me if you need any further information.

Sincerely, Hong Li, PhD Chinese Research Academy of Environmental Sciences Email: lihong@craes.org.cn 1. Please, update also the last section with atmospheric implications (see below)

The last section with atmospheric implications has been updated according to the suggestion of Reviewer 1. (Page 17, line 381-387)

"Based on the molecular composition detected in the mixed experiments, we propose a mechanism of high-molecular-weight compounds formation from the reaction of intermediate products originated from different precursors. This indicates that high-molecular-weight compounds (some of them are N- and/or S-containing species) in the ambient environment might be formed from the interactions of different precursors in the presence of NOx and SO2. When analysing the source of the detected aerosol species in the atmospheric environment, possible interactions from different VOC types need to be considered. In addition, the interactions between VOCs should be taken into account when evaluating the particle formation potential based on the monitored VOCs and oxidants."

 The statement "The data used in this study are available upon request from the corresponding author" needs to be changed. Namely, data should be available in the repositories. Please, check the manuscript preparation guidelines for authors. https://www.atmospheric-chemistry-and- physics.net/policies/data\_policy.html.

This has been changed. (Page 18, line 395-398)

"Data availability. All data supporting the conclusions of this paper are available either through the links provided below or upon request from the corresponding authors (lihong@craes.org.cn, gemaofa@iccas.ac.cn).

https://figshare.com/articles/dataset/\_acp-2022-294\_Effects\_of\_OH\_radical\_and\_SO2\_concentrati ons\_on\_photochemical\_reactions\_of\_mixed-anthropogenic\_organic\_gases/20437134"

**Technical comments/ errors:**

3. Line 36: Still not ok. Correct as: »Laboratory studies of long-chain alkanes, as representatives of IVOCs, are mainly focused on the case of a single long-chain alkanes or mixture of various precursors, which include long....

This has been corrected. (Page 2, line 36-38)

4. Line 42: The sentence needs to be corrected in a way: "For the mixture of various precursors including long-chain alkanes, studies are mainly focused on the chemical composition of the mixture gases, the properties of total organic carbon, the amount of SOA generated, ...

This has been corrected. (Page 2, line 42-44)

- Line 86: ...than in the ambient environment
   This has been added in the manuscript. (Page 3, line 85)
- 6. Line 90: Correct as: "... as the OH precursor; therefore, NO and HONO experiments were designed, as their pathways for generating OH radicals...."
  This has been corrected. (Page 3, line 89)
- 7. Line 102: "Gaseous NOx, SO2, and O3 concentrations inside the chamber were monitored in real time by an SO2 analyzer...« Please change as: »...by NOx (), ...SO2 () and O3 () analyzers.«
  This has been corrected. (Page 4, line 101-102)
- 8. Line 118/119: The sentence can be deleted; but "sulfate and nitrate" can be added in the parentheses after "in aerosols ()" and "nitric acid" after gases ().
  The sentence has been deleted, and the corresponding contents have been added to the specified location. (Page 4, line 114-115)
- 9. Lines 188-191: Can you write more understandable?

The figure caption has been modified. (Page 7, line 185-188)

"Each data point in this figure represents the average value (average OH concentration for Figure 1a, average OH exposure for Figure 1b) during the two sampling time periods. The point at 11:00 am represents the average value of the data during the period from reaction initiation to 11:00 am."

Line 232: Please correct: »to over predict the O3 concentration in the after 5.5 h«. (after 0.5 or 5.5h?)

This has been corrected in the sentence. (Page 9, line 229)

"to over predict the O3 concentration in the after 0.5 h".

11. Line 233: Correct the sentence: "This phenomenon is similar to a study about 1,3,5-TMB, the experiment of which was performed with an outdoor chamber"

This sentence has been corrected. (Page 9, line 229-231)

"Metzger et al. (2008) also reported the phenomenon that the model overpredicted ozone at a later stage of the 1,3,5-TMB degradation experiment, and the experiment was performed with an outdoor chamber."

12. Line 313: Correct to "Organic chemical composition" This has been corrected. (Page 13, line 311)

- 13. Line 347: ....larger concentration of organo-nitrates...This has been added in the sentence. (Page 15, line 345)
- 14. Line 358: Instead of organo-surfates should be organo-sulfates" This has been corrected. (Page 16, line 356)

**Please, revise/edit:**

15. Conclusions and atmospheric implications: Please revise/edit the last section with atmospheric implications. According to the suggestion of Reviewer 1, you constructed a mechanism explaining the experimental observations; but try (if possible) to use this mechanism in discussion and explanation of the atmospheric implications of your findings.

This has been added in Section 4. (Page 17, line 381-387)

"Based on the molecular composition detected in the mixed experiments, we propose a mechanism of high-molecular-weight compounds formation from the reaction of intermediate products originated from different precursors. This indicates that high-molecular-weight compounds (some of them are N- and/or S-containing species) in the ambient environment might be formed from the interactions of different precursors in the presence of NOx and SO2. When analysing the source of the detected aerosol species in the atmospheric environment, possible interactions from different VOC types need to be considered. In addition, the interactions between VOCs should be taken into account when evaluating the particle formation potential based on the monitored VOCs and oxidants."